# Influence of Insoluble Dietary Fibre on Expression of Pro-Inflammatory Marker Genes in Caecum, Ileal Morphology, Performance, and Foot Pad Dermatitis in Broiler

**DOI:** 10.3390/ani12162069

**Published:** 2022-08-14

**Authors:** Mariella Liebl, Martin Gierus, Christine Potthast, Karl Schedle

**Affiliations:** 1FFoQSI GmbH, Austrian Competence Centre for Feed and Food Quality, Safety and Innovation, 3430 Tulln, Austria; 2Department of Agrobiotechnology, Institute of Animal Nutrition, Livestock Products and Nutrition Physiology, University of Natural Resources and Life Sciences Vienna, Muthgasse 11, 1190 Vienna, Austria; 3Agromed Austria GmbH, 4550 Kremsmünster, Austria

**Keywords:** broiler, performance, insoluble dietary fibre, intestinal morphometrics, foot pad dermatitis, gut health

## Abstract

**Simple Summary:**

The objective of this study was to evaluate the effect of three different insoluble dietary fibre sources regarding their effect on growth performance, as well as foot pad dermatitis and physiological relevant parameters of the gastro-intestinal tract. The results showed that in the grower phase, supplementation of each fibre source improved body weights and weight gain. Moreover, the intestinal morphology and foot pad dermatitis were positively affected by moderate amounts of dietary fibre. Hence, this study indicates that the supplementation of moderate amounts of insoluble dietary fibre to diets, low in dietary fibre, triggers positive effects on foot pad dermatitis, ileal morphometrics, and specific performance parameters.

**Abstract:**

In a low-fibre diet destined for broilers, the effects of two lignocellulose products and soybean hulls were evaluated regarding their effect on ileal morphometric parameters, caecal gene expression, foot pad dermatitis, and performance. A total of 5040-day-old broilers (Ross 308) were allotted to four treatments and fattened for 36 days applying a three-phase feeding program. The control diet consisted of corn, wheat, and soybean meal. Experimental diets were supplemented with 0.8% lignocellulose product 1, 0.8% lignocellulose product 2, or 1.6% soybean hulls. Tissue samples for caecal expression of inflammation-related genes and ileal morphometries were collected on day 21. Gizzard pH and weights were recorded, and foot pad scores were evaluated at day of slaughter (day 36). In starter (day 1–10) and finisher phase (day 28–36), no effect on the performance was observed. In grower phase (day 11–27), fibre-supplemented diets showed significantly heavier body weights and daily weight gains (*p* < 0.05). Daily feed intake, feed conversion ratio, and gene expression analysis were unaffected by dietary fibre supplementation. Positive effects regarding ileal morphometrics (higher villi) and foot pad health occurred in fibre-supplemented diets. In conclusion, fibre supplementation improved performance in grower phase and showed beneficial effects regarding ileal morphology and foot pad dermatitis.

## 1. Introduction

In the feeding of livestock, different attempts to optimise the performance via promoting health are available. One strategy already well described in monogastric nutrition is the application of moderate amounts of materials high in dietary fibre. Primarily, fibre has been considered as an undesirable diluent or even an anti-nutritional factor in broiler feeding. However, the moderate supply of dietary fibre to chickens has undergone a paradigm shift and has attracted research interest, particularly with regard to animal welfare. For performance and welfare of the chicken, the maintenance of intestinal integrity and equilibrium are crucial and imply interactions between physical, chemical, physiological, and microbiological processes. Former studies indicate that depending on their physico-chemical properties, moderate amounts of fibrous components in feedstuffs for poultry may promote this equilibrium by positively affecting the development, functionality, and morphology of the gastrointestinal tract (GIT). As a result, the digestibility of nutrients as well as the performance of broilers can be improved. Moreover, moderate amounts of dietary fibre have shown to decrease litter moisture [1], the main culprit for foot pad dermatitis in boiler flocks. However, the response of the chicken to additional dietary fibre varies strongly with the amount and the source of the chosen fibre, as well as its physico-chemical properties. Soluble dietary fibre (SDF) tends to be easily fermented in a large part by the microbiota in the large intestine [2]. The microbial metabolic products of dietary fibre fermentation are short-chain fatty acids (SCFA). In addition to their function as an energy source, they also can trigger and ameliorate the immune status of the animal, and thus decrease the energetic effort for immune responses [3,4]. It was observed that changes in the caecal microbial population can be initiated when fermentable dietary fibre is supplemented [3]. Besides these well described positive effects, solubility of dietary fibre can enhance the digesta viscosity, which might hinder nutrient diffusion and absorption [5]. Furthermore, the solubility of dietary fibre mainly alters the gastrointestinal digesta in a physico-chemical way. On the other hand, the structural and physical properties of insoluble dietary fibre (IDF) affect the transit time of the digesta, as well as the gut motility by its inert characteristics when passing through the GIT [2]. Thus, changes in the morphological characteristics of the GIT, and therewith digestibility of nutrients, can occur, leading either to increased absorption surface [6] or epithelial abrasion [2]. Besides the inclusion rate, the lignification is also decisive for the mode of action of the insoluble fibre—if positive or negative [1]. A dietary fibre source very high in lignification is lignocellulose (Li), which due to precisely this fact has gained the interest of nutritionists. On the other hand, soybean hulls are characterised as a mainly insoluble low-lignified fibre source [7].

The objective of this study was to investigate the effect of the supplementation of moderate amounts of various insoluble dietary fibre sources, differing in their lignification, solubility, and fermentability characteristics, in a non-challenging feeding trial. The effects on performance traits, foot pad dermatitis, ileal morphology, and caecal gene expression of inflammation-related genes were evaluated. We hypothesised that the supplementation of moderate amounts of the insoluble dietary fibre sources to low-fibre diets beneficially promotes performance and foot pad health of broiler chickens. Furthermore, we hypothesised that these latter mentioned effects are more dominant with lignocellulose products, whereas the low lignified fibre sources have a more pronounced impact on the expression of pro-inflammatory cytokines genes in caecum.

## 2. Materials and Methods

### 2.1. Animals and Housing

As usual under practical conditions, a total of 5040 one-day-old mixed-sexed broiler chickens (Ross 308) were allotted to four dietary treatments in 36 pens in a climate-controlled house, resulting in nine replicates per treatment. Birds were reared to 36 days of age on a three-phase feeding program consisting of starter (day 1–day 10), grower (day 11–day 28), and finisher (day 29–day 36). Pens (8.2 m^2^) were equipped with manually filled tube feeders and nipple drinkers; new wood shavings were applied as litter. Initial temperature was set at 30 °C on day one and was continuously reduced to 20 °C on day 36; relative humidity was maintained between 60 and 65%. The study was conducted under compliance with the first regulation of keeping animals (BGBI. II no. 485/2004). All experimental procedures were approved by the Ethics Committee of the University of Natural Resources and Life Sciences, Vienna (reference number 2021/003). The study was carried out at the International Poultry Testing Station in Ústrašice, Czech Republic.

### 2.2. Diets and Feeding

Four diets were formulated, on the basis of maize, soybean meal, and wheat, to reach similar available nutrient content, as shown in Table 1. The diets were fed ad libitum in pelleted form. Treatment 1 (control C) acted as a control diet and was without further fibrous additive, resembling a commercial European maize-, wheat-, and soybean-based diet for broilers. Experimental diets either included 0.8% lignocellulose product 1 (Li1), 0.8% lignocellulose product 2 (Li2) according to the supplier recommendation, or 1.6% soybean hulls (SH) to reach a similar dietary fibre addition. Whereas Li1 is a standard lignocellulose, Li2 is declared as a lignocellulose product with increased polyphenol content and higher susceptibility to microbial fermentation. The reported values for the commercially available Li1 are total dietary fibre (TDF) = 87.6% FM, IDF/SDF 78.2; and for Li2: TDF = 85.3% FM, IDF/SDF 71.8; and for SH: TDF = 66.8 ± 5.2% FM, IDF/SDF 8.8 ± 0.4 [7].

### 2.3. Performane and Carcass Characteristics

Body weight (BW) was measured pen-wise on days 1 and 10 and individually on days 28 and 36. The average daily weight gains (ADG) were calculated for the three phases as well as for the overall experiment. Average daily feed intake (ADFI) was recorded pen-wise, and mortality corrected feed conversion ratio (FCR) was calculated for the periods 1–10 days, 11–28 days, 29–36 days, and for the overall experiment (1–36 days).

On day 36, 10 chickens per pen (two 
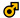
 and two 
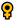
; n = 360), close to the pen’s average BW, were selected and applied for the analysis of carcass characteristics. Of these, four birds per pen (two 
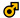
 and two 
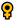
; n = 144) were used for gizzard examination. The chickens were slaughtered without prior feed restriction by electrically stunning and exsanguination. After removing the digestive tract, the gizzard weight was captured, and digesta pH was measured directly. The remaining birds were fasted for 12 h and killed the same way on the following day and subjected to carcass analysis. Regarding foot pad dermatitis (FPD), the feet of 30 birds per box (n = 1080) were examined, evaluated, and given a score according to the visual scale emerging out of the Welfare Quality^®^ Assessment protocol for poultry [9]. Score 0 is to be understood as “no evidence of FPD”, score 1 and 2 as “mild FPD”, and “score 3 and 4 as “severe FPD”.

### 2.4. Sample Collections and Analyses

Samples of every phase and diet were prepared and analysed according to the standard procedures [10] for dry matter (DM), crude fibre (CF), ether extract (EE), sugar, and starch. Crude protein concentration was calculated by determining nitrogen content using the Dumas combustion method (DuMaster 480, Büchi AG, Flawil, Switzerland) and by multiplying by 6.25 [11]. Gross energy content was analysed using bomb calorimetry (IKA C 200, IKA Werke GmbH and Co. KG, Staufen, Germany). Insoluble dietary fibre of the diets was analysed according to the AOAC method 991.43 and carried out with ANKOM Fibre Analyzer (ANKOM, Macedon, NY, USA). Values for SDF were beneath the detection level. On day 20 of the feeding trial, two birds per pen, corresponding to the average BW of the pen, were selected and applied for the intestinal sampling (one 
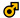
 and one 
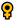
; n = 72). The wet sieving and particle size distribution was examined according to Röhe et al. (2014) [12] The un-fasted and un-plucked chickens were electrically stunned and scarified by exsanguination. After removing the digestive tract, the ileal section was dissected and gently rinsed with physiological saline solution. Samples from 2 cm cranial to the ileocaecal junction were excised and stored in a 10% buffered formaldehyde solution. After 24 h of dehydration with ethanol (70%), the samples were drained, paraffin embedded, sectioned via rotary microtome (5 µm; Leica RM 2255, Leica Biosystems GmbH, Nussloch, Germany), and stained (Leica Auto Stainer XL ST5010, Leica Biosystems GmbH, Nussloch, Germany) following the protocol with Alcian blue and periodic acid–Schiff. Six well-oriented, representative villi and crypts as well as muscular layers of each sampled animal were examined with computerised light microscopy (Leica DM 600 B, Darmstadt, Germany) using the Leica Application Suit software (Leica, Version 4.12). Caecal samples were collected for the evaluation of the gene expression of the inflammation-related genes. Gene expression analyses were carried out to reach the MIQE guidelines [13]. Samples were rinsed with physiological sodium chloride solution, embedded in cryovials (2 mL, steril, Biozym Scientific GmbH, Oldendorf, Germany) immediately, frozen in liquid nitrogen, and stored at −80 °C until RNA extraction, which was carried out with TRI Reagent© solution according to the manufacturer’s protocol (Sigma Aldrich, Steinheim, Germany). To quantify concentration of each extracted RNA, a spectral analysis was carried out using a NanoDrop (ND-100 Spectrophotometer, Thermo Fisher Scientific, Waltham, MA, USA), whereas the RNA integrity was determined with a chip-based electrophoresis system (Experion^TM^ Automated Electrophoresis System, Bio-Rad, Hercules, CA, USA). The total RNA was reverse-transcribed with a QuantiTect Rev Transcription Kit (Qiagen, Hilden, Germany), following the manufacturer’s instructions. PCR was performed for each gene measured in triplicate on Rotor-Gene Q (Qiagen, Hilden, Germany) with SYBR Green PCR Kit (Qiagen, Hilden, Germany), following the manufacturer´s instructions. The reference genes used for normalisation in the quantitative real-time PCR (qPCR) were *glyceraldehyde-3-phosphate dehydrogenase, Actin-β*, and *ubiquitin*; target genes were *interleukin 1β (IL1β)*, *interleukin 6 (IL6)*, *interleukin 8 (IL8)*, *tumour necrosis factor alpha (TNF-*α), and *nuclear factor kappa B (NF-κ*B). All primers were obtained from Eurofins Genomic, taken out of previous studies [3,14,15,16] and individually tested. Gene expression was calculated using the ΔΔCT method [17]. The characteristics of the primers are displayed in Table 2.

### 2.5. Statistic Analyses

Statistical analysis of the gathered data was conducted using either a mixed procedure (PROC MIXED) for parametric values combined with Tukey–Kramer post hoc test or the Mann–Whitney-Wilcoxon test for nonparametric values (PROC NPA1WAY) of the Statistical Analysis Software (SAS ^®^ version 9.4., Cary, CA, USA). Treatment groups were stated as fixed factor. For performance parameters and molecular analyses, chicken pens were considered as an experimental unit and animals for carcass characteristics and morphometric parameters. For carcass characteristics, the sex of the broiler was included in the model. Significance for statistical tests was set at *p* < 0.05.

## 3. Results

### 3.1. Performance and Carcass Characteristics

The performance data showed significant differences in the grower phase only (Table 3). Here, the birds fed SH and Li1 increased ADG compared to the C (*p* < 0.05). This resulted in significant heavier BW at day 28 regarding the treatments SH and Li1 compared to the C and a tendency towards heavier BW with Li2. Chickens included Li1 in the diet improved FCR in a trend compared to C (−2.7%, *p* = 0.09).

No differences regarding slaughter or eviscerated carcass weights were recorded (*p* > 0.1). Gizzard weights as well as pH remained unaffected by the dietary fibre supplementation (*p* < 0.05). Carcass analyses revealed significant differences in abdominal fat content and leg weights (Table 4). Broilers fed Li1 showed the highest abdominal fat content and a significant difference to birds fed C. Furthermore, Li1 and Li2 reduced the leg weight significantly when compared to C (*p* < 0.05). Significant higher values for male birds were observed regarding body and eviscerated carcass weight, as well as for abdominal fat, wings, legs, breast, and gizzard percentage (*p* < 0.05).

### 3.2. Foot Pad Health

The highest FPD score was determined in the control group (Figure 1). Here, 43% of the broilers had a score 3 or 4 in contrast to the lowest scores obtained with Li1 where only 22% showed these values. In the control treatment, the statistical analysis further revealed significant higher FPD scores compared to the experimental treatments with additional dietary fibre (*p* < 0.001).

### 3.3. Ileal Morphology and Caecal Inflammatory Cytokine Gene Expression

Dietary fibre addition with either lignocellulose (Li1, Li2) or SH increased the heights of the ileal villi significantly compared to the control treatment (*p* < 0.050; Table 5). Villus area significantly increased with Li1 compared to the C treatment (*p* = 0.045). Significant deeper crypts were measured in birds fed SH and Li1 (192.93 and 189.22 µm) as well a strong trend towards deeper crypts with Li2 (177.99 µm; *p* = 0.0547). Regarding the muscularis layer, Li2 increased the thickness significantly when compared to C (*p* < 0.05). Goblet cell count per 100 µm of villus height and per 100 µm of crypt depth, as well as the VH/CD ratio, were similar in all treatments. Overall, in birds receiving the control treatment, the thinnest intestinal layers were recorded.

The results of the evaluation of the inflammation-related caecal gene expression of the experimental diets are summarised in Table 6. Although various up- and downregulations of genes associated with immune response occurred, no consistent or statistically relevant effects of the fibre sources were detected in comparison to the control treatment (*p* > 0.10). All samples taken were within the normal physiological scope.

## 4. Discussion

### 4.1. Performance and Carcass Characteristics

It is well known that diets with strong increasing contents of dietary fibre reduce nutrient digestibility and as a result performance if diets were not balanced for their available nutrient content [18,19,20]. Moreover, high amounts of Li (supplementation level 5% and 10% without nutrient balancing) can dilute nutrient content and affect a marked decrease in ADG, while feed intake tends to increase simultaneously [21]. On the other hand, reports of the supplementation with moderate amounts of dietary fibre in diets for broilers range from none to performance-improving effects [20,22]. The present study focused on different sources of mainly insoluble fibre in broiler feeding, which showed significant positive effects on performance in the grower phase of fattening. Interestingly, the sieve analysis showed a decrease in the coarse grinded (>1 mm diameter) and an increase in the fine grinded fraction (<0.5 mm diameter) in diets supplemented with dietary fibre. The reason for this phenomenon is difficult to explain; however, it is well known that a finer particle size can increase digestibility and as a result improve performance [23]. Similar results regarding performance were observed by González-Alvarado et al. (2007) [24] showing enhanced BWG and FI, and improved FCR from day 14 to 21 with the supplementation of 3% soybean or oat hulls in a corn-based control diet containing 2.5% crude fibre. Regarding the whole fattening period (day 1 to 21), the inclusion of oat or soybean hulls improved BWG and FCR without effecting FI. The authors justified the performance-improving effects by an enhanced gizzard function and as a result an increased nutrient digestibility affected by the moderate fibre supplementation [20]. However, these results are in contrast to the study of Makivic et al. (2019) [25] where improving effects were already visible from the starter phase, supplementing 0.6% Li to their diet. In the present study as well as in the study of Makivic et al. (2019) [25], a similar diet was applied. However, the study designs differ regarding genetics (Ross 308 vs. Cobb 500), litter, and housing between these two studies. In contrast, Zeitz et al. (2019) [3] observed no effects on performance when applying 0.8% of two different Li sources. Besides the different breeding lines and examined gut sections, the major difference between the study of Zeitz et al. (2019) [3] and our study can be considered in the diet formulation and their chemical composition. The diets of Zeitz et al. (2019) [3] as well as our own diets were based on corn, soybean meal, and wheat. However, they were in different ratios (1.4:0.9:1 vs. 3.1:1.4:1), resulting in crude fibre content being approximately twice that of our diets. Hence, differences concerning the diet formulation make it hard to draw distinctions between moderate dietary fibre- and formulation-related impacts or management strategies. This opinion is supported by several studies examined in the last decade, and is also the main statement in the conclusion of the review from Röhe and Zentek (2021) [22]. The authors stated that the composition has a greater impact on BW of broilers than Li [22]. Results obtained in broiler trials applying a comparable Li concentration to our study showed contradictory results. The authors reported from no effects up to enhanced dFI and ADG as well as an improved FCR affected by moderate Li supplementation [22]. Regarding feed intake, no trend could be recognised whether increasing or decreasing. It is probable that the level and type of dietary fibre, as well the age of the bird, modifies the response of poultry concerning FI and FCR [24]. The importance of taking the diet formulation into account was clearly shown in a feeding trial by Makivic et al. (2019) [25]. In their study, 0.6% of a Li product was added once at the expense of 0.3% soybean meal and 0.3% maize, and second at the expense of 0.6% soybean meal only, resulting in significant differences concerning BW. Thus, it is obvious that even minimal deviations in matrix formulation, as reported by Makivic et al. (2019) [25], can affect results and reduce the comparability with other studies feeding other feed matrixes.

The carcass analysis showed no statistical effect on the slaughter weight or eviscerated carcass. The data are similar to that of Zeitz et al. (2019) [3], who reported that additional partly fermentable Li showed no effect on the carcass traits of male Cobb 500 broilers; however, in their study, dressing percentage increased up to two percentage points through the addition of 0.8% partly fermentable Li. Zeitz et al. (2019) [3] explained this with the reduction of pro-inflammatory cytokine gene expression found in the jejunum initiated by the fermentability of the Li. However, in context with the similar content of metabolisable energy and available nutrients among the treatments of this study, these effects were not visible and are difficult to explain. Depending on the quantity and quality of the chosen dietary fibre, interference with the digestion of nutrients can occur, influenced by their availability in the lumen [2]. Therefore, it can be presumed that the variations in solubility and fermentability of the additional dietary fibre in our study may have interfered with nutrient availability, resulting in differences among abdominal fat and leg weight.

In the present study, the well-described sexual dimorphism between female and male broilers showed a significant effect on several performance data and carcass characteristics [26].

Fibre may stimulate proventriculus HCl secretion and gizzard activity, resulting in lower pH values, longer retention time, and increased reflux of the digesta. Pepsin activity and subsequently the proteolytic degradation of proteins may be increased [27]. Concerning the effectiveness of insoluble dietary fibre, the particle size is of importance regarding the influence on the gizzard and its function as particle size reducer in degrading nutrients or as pacemaker for the whole GIT [1]. Course grinded insoluble fibre is likely to accumulate in the upper intestinal tract, thereby stimulating enzyme secretion and activity [28]. The gizzard operates to grind down particles over 500 µm [29]. However, in the present study, dietary fibre neither affected the relative gizzard weight nor the pH values (*p* > 0.100). Besides the low inclusion rates, this might have been due to the fact that, already upon entering the beak of the bird, the added lignocelluloses had a mean particle size below 200 µm, making the term “structural component” ineligible especially in the starter and finisher phase (Table 1).

### 4.2. Foot Pad Health

The solubility of dietary fibre sources is often mentioned in the context of excreta dry matter [30], which is the major culprit for feet health related issues of broilers [31]. According to Slama et al. (2019) [7], the physico-chemical properties of soybean hulls, Li1 and Li2, vary strongly when confronted with fluids. The application of Li products showed an approximately 10 times higher IDF to SDF ratio compared to SH [7]. Therefore, the variation in solubility of the applied fibre sources in our trial led us to the expectation that variation in excreta dry matter will occur. However, the dry matter remained unaffected (not displayed). This phenomenon might be explained by the low inclusion rates of the additional dietary fibre. Nevertheless, this study showed a significant alleviating effect of fibre supplementation on the FPD that might be interpreted as a result of the morphological findings, discussed below. Nevertheless, more research is necessary to clarify this mode of action.

### 4.3. Ileal Morphology

In poultry, a strong link between intestinal dysbiosis, epithelial barrier function, and inflammation-related activities exists. Pro-inflammatory cytokines can interact with epithelial cells, influencing the intestinal architecture and reducing the permeability for nutrients [32]. In this study, the variations in the intestinal architecture are clearly related to the added dietary fibre. An intact intestinal morphology can be seen as an indicator of intestinal health of the small intestine, with long villi and short crypts being considered a sign of a healthy, well-functioning gut mucosa. Short villi and deep crypts are associated with increased cell apoptosis and proliferation, which consequently elevates the expenditure of energy and proteins [32]. There is a common assumption that dietary fibre, and especially the insoluble fractions, have an abrasive effect on the intestinal lining [2], resulting in damaged epithelial structure showing atrophic short villi and thus endogenous loss enhances [1]. However, contradicting assumptions and results can also be found, describing a stimulation effect of insoluble fibre on cell proliferation with enhanced villus height and absorption area [33]. In this study, each insoluble fibre led to significant higher villi compared to the control treatment. Interestingly, the numeric highest villi and lowest crypts were recorded in the soybean hull treatment, including higher amounts of soluble dietary fibre fractions. Different effects of soybean hulls compared with pure insoluble cellulose regarding the intestinal morphology were also observed by Tejeda and Kim (2020) [34] who are explaining this effect with the higher amount of pectin in the hulls.

Bogusławska-Tryk et al. (2020) [35] reported that broilers fed diets with 0.5 and 1.0% additional Li formed significant higher but narrower ileal villus, resulting in unaffected size of the surface area. The crypt depth was decreased with Li addition, leading to the highest VH/CD ratio in treatments with 1.0% Li. Within the present study, the morphometric measurements revealed contradictory results. Although the supplemented dietary fibre also led to increased villi lengths, the crypt depth in our study was also increased, which resulted in an unaffected VH/CD ratio. Higher villus and deeper crypts might be a sign of a compensation strategy. An increased cell proliferation in the crypt consumes more energy; however, this effect might also be translated into higher villus with greater absorption efficiency [36]. Although VH/CD was unaffected, the significant higher villi and at least numerically increased nutrient absorption area might be a reason for the increased BW, with daily weight gain for treatments supplemented with dietary fibre. An increased density of goblet cells may also be indicative of enhanced mucus production through either surface abrasion by structural components in the digesta or an increased immunologic response. However, the unaffected goblet cell counts, both in villi and crypts, and the longer villi in the experimental treatments of this study, reveal that opposite effect occurred through the added dietary fibre sources.

Despite the variation in physico-chemical properties of the applied dietary fibre sources, no precise effects related thereto can be detected within this study design.

### 4.4. Caecal Inflammatory Cytokines Genes

The results of the gene expression analysis in this study are in accordance with the histometric measurements. Both show that in none of the treatments immune-relevant processes occurred or, more precisely, were necessary in this non-challenging experiment. These findings compare favourably with the study of Zeitz et al. (2019) [3], who also examined the gene expression in the ceca of broilers. In accordance with our results, *IL1ß*, *IL6*, and *IL8* were unaffected by either lignocellulose in their study. However, in contrast to our results, diets with 0.8% of a fermentable lignocellulose resulted in higher *TNF* expression compared with 0.8% of a standard lignocellulose product. Furthermore, Zeitz et al. (2019) [3] stated that low levels of 0.8% insoluble but fermentable lignocellulose can also change the caecal SCFA towards enhanced acetate and decreased butyrate contents, whereas standard lignocellulose without fermentable fractions showed no effect. In the present study, the expression of *NF-κB* in the ceca was not affected by the treatments. The expression of *NF-κB*, a transcription factor of cytokines, can be triggered by the presence of butyrate. Hence, it can be speculated that also the butyrate production in the caeca may have not been influenced by dietary treatments.

## 5. Conclusions

The data reveal a growth-promoting effect in grower phase when either lignocellulose products or soybean hulls are added as dietary fibre to a low-fibre diet. Each applied insoluble dietary fibre source, regardless of its physico-chemical properties, modified the ileal morphology and resulted in higher villi and deeper crypts. The prevalence of foot pad dermatitis was reduced by the additional dietary fibre. The examined parameters regarding the caecal immune response remained unaffected by the dietary fibre supplementation.

Therefore, it can be concluded that moderate amounts of insoluble dietary fibre supplementation, under the precondition of a nutrient balanced diet, have the potential to improve growth performance, ileal morphometrics, and foot pad dermatitis.

## Figures and Tables

**Figure 1 animals-12-02069-f001:**
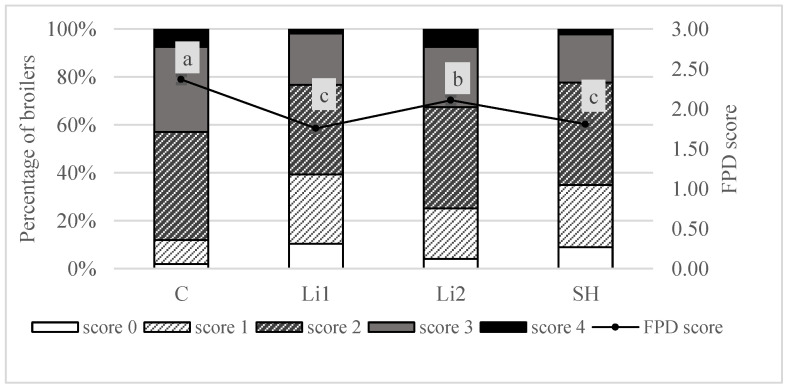
Percentage of affected broilers per foot pad dermatitis (FPD) score and average FPD score of broilers fed diets supplemented with either 0.8% lignocellulose product 1 (Li1), 0.8% lignocellulose product 2 (Li2), 1.6% soybean hulls (SH), or no special fibre source (C); ^a, b, c^ Values with different superscripts differ significantly (*p* < 0.05).

**Table 1 animals-12-02069-t001:** Ingredients and analysed nutrient composition of diets, as fed.

	Starter				Grower				Finisher			
Ingredients % FM	C	Li1	Li2	SH	C	Li1	Li2	SH	C	Li1	Li2	SH
Corn	50.0	49.6	49.6	49.0	50.0	49.6	49.6	48.9	52.5	52.1	52.1	51.4
Soybean meal, hp	28.6	28.4	28.4	28.0	27.8	27.6	27.6	27.3	23.6	23.4	23.4	23.1
Wheat	12.2	12.1	12.1	11.9	14.4	14.3	14.3	14.1	16.7	16.5	16.5	16.3
Corn gluten	2.5	2.5	2.5	2.5								
Plant-based oil	2.2	2.2	2.2	2.6	4.0	4.0	4.0	4.4	3.9	3.9	3.9	4.3
Premix ^1^	1.2	1.2	1.2	1.2	1.0	1.0	1.0	1.0	1.0	1.0	1.0	1.0
Monocalcium phosphate	1.1	1.1	1.1	1.0	0.8	0.8	0.8	0.8	0.6	0.6	0.6	0.6
Calcium carbonate	0.9	0.9	0.9	0.9	0.8	0.8	0.8	0.8	0.9	0.9	0.9	0.8
DL-methionine	0.3	0.3	0.3	0.3	0.3	0.3	0.3	0.3	0.3	0.3	0.3	0.2
Zootechnical additives ^2^	0.3	0.2	0.2	0.2	0.2	0.2	0.2	0.2	0.2	0.2	0.2	0.2
L-Lysin	0.2	0.2	0.2	0.2	0.1	0.1	0.1	0.1	0.1	0.1	0.1	0.1
Sodium chloride	0.2	0.2	0.2	0.2	0.2	0.2	0.2	0.2	0.2	0.2	0.2	0.2
Coccidiostats ^3^	0.2	0.2	0.2	0.2	0.2	0.2	0.2	0.2				
L-Threonine	0.1	0.1	0.1	0.1	0.1	0.1	0.1	0.1	0.1	0.1	0.1	0.1
Lignocellulose product I		0.8				0.8				0.8		
Lignocellulose product II			0.8				0.8				0.8	
Soybean hulls				1.6				1.6				1.6
Chemical composition %FM												
Dry matter	90.1	90.1	90.4	89.9	89.0	89.6	88.6	88.9	88.7	89.0	88.4	88.3
Crude protein	21.2	21.4	22.0	21.1	19.4	19.0	19.4	18.8	18.9	18.4	19.2	18.1
Ether extract	4.6	4.5	4.5	4.5	6.3	6.3	6.1	6.7	6.3	6.3	6.0	6.6
Crude fibre	2.6	2.4	2.7	2.6	1.8	2.2	2.0	2.2	2.3	2.1	2.4	2.2
IDF	13.5	13.9	13.8	14	13.3	13.5	13.5	14.1	11.6	11.7	12.1	11.8
SDF	>1	>1	>1	>1	>1	>1	>1	>1	>1	>1	>1	>1
Crude ash	5.5	5.3	5.4	5.3	5.2	4.7	4.6	4.8	4.7	4.4	4.5	4.2
Starch	42.8	43.4	42.9	43.2	43.3	43.9	43.0	42.7	44.1	45.6	44.0	45.1
Sugar	4.5	4.3	4.6	4.6	4.4	4.4	4.5	4.2	4.1	4.0	4.0	4.1
Gross energy MJ/kg	17.2	17.1	17.3	17.2	17.2	17.5	17.2	17.6	17.4	17.5	17.3	17.4
Calculated composition												
AME_N_ MJ/kg ^4^	12.6	12.7	12.7	12.6	13.0	13.0	12.9	12.9	13.0	13.1	12.9	13.1
Wet sieve analysis												
>1 mm, %	9.9	6.2	9.2	7.5	19.5	17.6	13.7	12.7	9.8	11.4	9.8	8.4
≥0.5–≤ 1 mm, %	30.9	30.6	33.3	34.8	34.1	31.4	31.5	32.9	35.7	37.1	35.4	36.2
<0.5 mm, %	59.2	63.2	57.5	57.8	46.4	51.1	54.8	54.5	54.6	51.2	54.8	55.4
dMean, mm	0.6	0.5	0.6	0.5	0.8	0.8	0.7	0.7	0.6	0.7	0.6	0.6

Abbreviations: C, no special fibre addition; Li1, standard lignocellulose; Li2, partly fermentable lignocellulose; SH, soybean hulls; FM, fresh matter; AMEN, apparent metabolisable energy nitrogen corrected. ^1^ Starter: 12,000 IU vitamin A, 5000 IU vitamin D3, 75 mg vitamin E, 50 mg Fe, 12 mg Cu, 70 mg Zn, 99 mg Mn, 1.5 mg J, 0.4 mg Se; grower and finisher: 10,000 IU vitamin A, 5000 IU vitamin D3, 75 mg vitamin E, 50 mg Fe, 12 mg Cu, 69 mg Zn, 99 mg Mn, 1.5 mg J, 0.4 mg Se; ^2^ 500 U phytate, 3.000 U xylanase, 675 U glucanase; ^3^ in starter: 40 mg Nicarbazin, 40 mg Narasin, in grower: 60 mg Salinomycin; ^4^ calculated according to the formula for compound feed of the society of nutrition physiology [8].

**Table 2 animals-12-02069-t002:** Characteristics of the primers used in this study.

Gene	Sequence 5′ to 3′		Product Length, bp	Annealing Temperature, °C	NCBI Accession
*GAPDH*	for.	GGTGGTGCTAAGCGTGTTAT	264	57.3	K01458
	rev.	ACCTCTGTCATCTCTCCACA		57.3	
*ACTB*	for.	ATGAAGCCCAGAGCAAAAGA	223	55.3	NM_205518
	rev.	GGGGTGTTGAAGGTCTCAAA		57.3	
*Ubiquitin*	for.	GGGATGCAGATCTTCGTGAAA	147	57.9	M11100
	rev.	CTT GCC AGC AAA GAT CAA CCT T		58.4	
*IL 1β*	for.	CATTACCGTCCCGTTGCTTT	105	57.3	NM 204524.1
	rev.	AGTCACAATAAATACCTCCACCC		58.9	
*IL6*	for.	CCAGAAATCCCTCCTCGCCAATC	222	64.2	NM 204628.1
	rev.	TGAAACGGAACAACACTGCCATC		60.6	
*IL8*	for.	TGCTGTGGGATTCACTGTCCA	93	59.8	HM179639.1
	rev.	ACTGAAGTGGCTTCCAAGGGA		59.8	
*TNF-*α	for.	CAGGACAGCCTATGCCAACA	95	59.4	NM 204628.1
	rev.	CATCTGAACTGGGCGGTCAT		59.4	
*NF-kB*	for.	GAAGGAATCGTACCGGGAACA	131	59.8	NM_205134.1
	rev.	CTCAGAGGGCCTTGTGACAGTAA		62.4	

Abbreviations: *GAPDH*, *glyceraldehyde-3-phosphate dehydrogenase*; *ACTB*, *Actin-β*; *IL1β*, *interleukin 1 beta*; *IL6, interleukin 6*; *IL8*, *interleukin 8*; *TNF-α*, *tumour necrosis factor alpha*; *NF-κB*, *nuclear factor kappa beta*.

**Table 3 animals-12-02069-t003:** Growth performance.

	C	Li1	Li2	SH	SEM	*p*-Value
Animals day 1, n	1260	1260	1260	1260		
Animals day 10, n	1260	1260	1260	1260		
Animals day 28, n	1229	1229	1230	1231		
Animals day 36, n	1223	1225	1229	1226		
Ratio male/female at day 36, n	610/613	616/609	618/611	625/601		
Body weight (BW), g						
Day 1 ^1^	39	39	39	39	0.1	0.994
Day 10 ^1^	240	243	242	244	1.3	0.791
Day 28 ^2^	1436 ^b^	1517 ^a^	1490 ^a^	1498 ^a^	9.0	0.005
Day 36 ^3^	2028	2083	2095	2109	13.7	0.178
Average daily weight gain ^1^ (ADG), g						
Starter	20	20	20	21	0.1	0.785
Grower	66 ^b^	70 ^a^	69 ^ab^	69 ^a^	0.5	0.006
Finisher	85	81	87	87	1.6	0.541
All	57	58	59	59	0.4	0.176
Daily feed intake ^1^ (dFI), g						
Starter	27	27	27	27	0.2	0.906
Grower	97	99	98	98	0.5	0.777
Finisher	170	170	171	172	2.0	0.990
All	92	93	93	93	0.6	0.955
Feed conversion rate ^1^ (FCR), kg/kg						
Starter	1.35	1.35	1.35	1.32	0.01	0.686
Grower	1.48	1.41	1.43	1.43	0.01	0.101
Finisher	2.02	2.12	2.00	1.99	0.03	0.533
All	1.61	1.57	1.57	1.56	0.01	0.368

Abbreviations: C, no special fibre addition; Li1, standard lignocellulose; Li2, partly fermentable lignocellulose; SH, soybean hulls; SEM, standard error of mean. ^a, b^ Values with different superscripts differ significantly (*p* < 0.05); ^1^ n = 9 pens/treatment; ^2^ n = 4919 birds in total; ^3^ n = 4903 birds in total.

**Table 4 animals-12-02069-t004:** Carcass characteristics of broiler chickens.

	C	Li1	Li2	SH	SEM	*p*-Value	
						Diet	Sex
Body weight (BW) ^1^, g	2171	2249	2220	2222	13	0.177	<0.001
Eviscerated carcass ^1,^ g	1753	1815	1792	1795	10	0.173	<0.001
% Eviscerated carcass ^1^							
Abdominal fat	0.91 ^b^	1.05 ^a^	0.94 ^ab^	1.00 ^ab^	0.02	0.004	<0.001
Heart	0.64	0.62	0.59	0.61	0.01	0.133	0.569
Liver	2.40	2.40	2.39	2.41	0.02	0.962	0.683
Head and neck	4.80	4.75	4.83	4.85	0.03	0.644	0.638
Wings	8.98	8.93	8.98	8.93	0.03	0.881	0.001
Legs	4.34 ^a^	4.03 ^b^	4.13 ^b^	4.18 ^ab^	0.03	0.001	<0.001
Breast	29.54	29.39	29.71	29.31	0.09	0.408	0.047
Thigh	26.00	25.79	26.18	26.15	0.07	0.204	0.522
Gizzard ^2^, g/kg BW	9.73	10.09	9.44	9.69	0.16	0.562	<0.001
pH gizzard ^2^	2.95	3.02	2.98	3.10	0.05	0.769	0.485

Abbreviations: C, no special fibre addition; Li1, standard lignocellulose; Li2, partly fermentable lignocellulose; SH, soybean hulls; SEM, standard error of mean. ^a, b^ Values with different superscripts differ significantly (*p* < 0.05); ^1^ n = 90 birds/treatment; ^2^ n = 36 bird/treatment.

**Table 5 animals-12-02069-t005:** Ileal morphological characteristics ^1^.

		C	Li1	Li2	SH	SEM	*p*-Value
Villus							
Height	µm	671.07 ^b^	742.35 ^a^	724.00 ^a^	753.38 ^a^	6.75	0.001
Villus area calculated ^2^	mm²	0.32 ^b^	0.38 ^a^	0.34 ^ab^	0.35 ^ab^	0.01	0.053
Goblet cells	n/100 µm villus height	19.75	20.98	19.64	17.69	0.47	0.149
Crypt							
Depth	µm	160.00 ^b^	189.22 ^a^	177.99 ^a^	192.93 ^a^	2.56	0.001
Goblet cells	n/100 µm crypt depth	20.83	22.98	22.83	20.64	0.54	0.290
VH/CD		4.41	4.24	4.48	4.26	0.07	0.552
Muscularis	µm	170.88 ^b^	168.00 ^b^	185.79 ^a^	179.08 ^ab^	1.93	0.002

Abbreviations: C, no special fibre addition; Li1, standard lignocellulose; Li2, partly fermentable lignocellulose; SH, soybean hulls; SEM, standard error of mean; VH/CD, villus height/crypt depth; ^1^ n = 18 birds/treatment; ^2^ area = 2π × villus width/2 × villus height; ^a, b^ values with different superscripts differ significantly (*p* < 0.05).

**Table 6 animals-12-02069-t006:** Relative caecal gene expression ^1^ (group C = 1).

	C	Li1	Li2	SH	SEM	*p*-Value
*IL 1β*	1.00	0.76	1.49	1.93	0.24	0.337
*IL6*	1.00	1.38	0.99	0.99	0.14	0.716
*IL8*	1.00	0.74	1.56	1.11	0.27	0.619
*TNF-α*	1.00	0.77	1.84	1.63	0.25	0.392
*NF-kB*	1.00	0.88	0.74	0.86	0.12	0.907

Abbreviations: C, no special fibre addition; Li1, standard lignocellulose; Li2, partly fermentable lignocellulose; SH, soybean hulls; SEM, standard error of mean; *IL1β*, *interleukin 1 beta*; *IL6*, *interleukin 6*; *IL8*, *interleukin 8*; *TNF-*α, *tumour necrosis factor alpha*; *NF-κB*, *nuclear factor kappa beta*; ^1^ n = 18 birds/treatment.

## Data Availability

The data presented in this study are available on request from the corresponding author.

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
