# Peer review of "Influence of Insoluble Dietary Fibre on Expression of Pro-Inflammatory Marker Genes in Caecum, Ileal Morphology, Performance, and Foot Pad Dermatitis in Broiler"

_animals, 2022, doi:10.3390/ani12162069_

Round 1

Reviewer 1 Report

The paper is written ok. I recommend the manuscript for publication after minor revision considering the following comments:

Although the work is of potential interest and valuable to the reader, the manuscript suffers some flaws and needs editing, here are some general comments:

1)     In your introduction, the research background could be improved. It should add some gut microbiota research results related to insoluble fibre. Please reorganize

2)     In your result, why you focus on caecal while not jejunal and ileal inflammatory factors. Usually, the level of inflammatory cytokines in jejunum or ileum are more valuable for immune response. Please explain

3) In your discussion, please elaborate a sentence highlighting benefits of insoluble fibre to chickens.

Minor comments (examples rather than an exhaustive list of corrections):

Title:

Please indicate what kind of genes.

Keywords:

Too many keywords, please limit the number to five.

 Introduction

Line 84: please change “one the other hand” to “on the other hand”

Line 98: “on caecal gene expression” is an incomplete sentence, please indicate what kind of genes.

Materials and Methods

Table 1: Please put the level of Lignocellulose product I and Lignocellulose product II in the right line. Please check.

Line 213: please change “ΔΔCP” to “ΔΔCT

Results:

3.3 Ileal morphology and gene expression

Please complete the subtitle, for example, “Ileal morphology and caecal inflammatory cytokines genes expression”.

Discussion:

4.1 Perfomance and carcass characteritsics

It would be better to describe that insoluble dietary fibre changed the structure of gut microbiota and is associated with improved growth performance from some references.

4.4 Gene expression

Please complete the subtitle, for example, “Caecal inflammatory cytokines genes expression”.

Line 476: it is better changing “must” to “may”.

Conclusion:

This part summarized not well, the last sentence was too long.

Author Response

Dear reviewer, thank you for your valuable comments. In the modified manuscript we have carefully incorporated your suggestions. We hope the modified manuscript is now able for publication.

Best regards

Karl Schedle

Reviewer 2 Report

1.      The simple summary is suggested to rewrite to briefly describe aims, major findings, and conclusions instead of looking like introduction.

2.      In introduction, the authors need to provide some clues why dietary fiber is hypothesized to affect foot pad health/dermatitis. It is rather difficult for readers to link the two independent concepts to cut into the manuscript. Also, in discussion, the authors need molecular and/or physiological evidences

3.      The authors need to rationalize diet composition, particularly in CP%. In the starter, grower, and finisher, the highest and lowest crude protein is 22.0 and 21.1%, 19.4 and 18.8%, and 18.1 19.2%. How the authors concluded the results are due to the effect of fiber level instead of CP%? Also, ME levels varied despite very little.

The authentic avian (gallus) TNF-a was recently cloned (Rohde, et al., Front Immunol. 2018; 9: 605. ). How the authors validated the clone used for designing the primer sequences for qRT-PCR study was real gallus TNF-a?

Author Response

(The authors gave the same response as above.)

Reviewer 3 Report

In general, the concept of this work is interesting. But there are some points of criticism which need to be clarified before the publication.

Line 121, 222 – please write ad libitum as well as post hoc in italic.

Throughout the text – as a general rule, if the authors refer to genes, they should be written in italic.

Line 121 – correct to “Treatment 1 (control C) acted “

Line 124  – Was (and how) the control diet (commercial one) checked for the presence of fibrous additive?

Line 190 – Were caecal samples dissected unilaterally or bilaterally ? From histological point of view, the ileum/cecum are organs (not a tissue).

Line 191 – how many animals were subjected to genes expression analysis?

Line 210 – please correct to “tumor”.

Line 272 – photographic documentation of the observed morphological changes are missing.

Figure 1 – what (c) superscript stands for?

Line 293 – what “expression1” means?

Line 479 – The conclusions are rather enigmatic. What is the meaning of “positive” or “beneficial” (in what sense?). The term “food-pad health” is awkward.

Author Response

(The authors gave the same response as above.)
